# Naturally Occurring Hepatitis B Virus Mutations Leading to Endoplasmic Reticulum Stress and Their Contribution to the Progression of Hepatocellular Carcinoma

**DOI:** 10.3390/ijms20030597

**Published:** 2019-01-30

**Authors:** Yu-Min Choi, So-Young Lee, Bum-Joon Kim

**Affiliations:** Department of Microbiology and Immunology, Biomedical Sciences, Liver Research Institute and Cancer Research Institute, College of Medicine, Seoul National University, Seoul 03080, Korea; cym486486@gmail.com (Y.-M.C.); leesy8822@snu.ac.kr (S.-Y.L.)

**Keywords:** Hepatitis B virus infection (HBV), Endoplasmic Reticulum stress, HBV surface antigen (HBsAg), preS/S mutations, core mutations, HBx mutations, Unfolded protein response (UPR), PERK, IRE1, ATF6, Hepatocellular carcinoma (HCC), apoptosis

## Abstract

Hepatitis B virus (HBV) infection is a global health problem that causes a wide range of pathological outcomes, including cirrhosis and hepatocellular carcinoma (HCC). Endoplasmic reticulum (ER) stress induction by HBV infection has been implicated in liver carcinogenesis and disease progression with chronic inflammation via enhanced inflammation, oxidative stress-mediated DNA damage, and hepatocyte proliferation. In the natural course of HBV infection, the accumulation of naturally occurring mutations in the HBV genome can generate several mutant types of HBV-encoded proteins, including three different proteins in the S ORF (SHBs, MHBs, and LHBs) and HBcAg in the C ORF, which could contribute to enhanced ER stress in infected hepatocytes mainly via increased ER accumulation of mutant proteins. However, it seems that there may be distinct capacity and pathway in ER stress-induction and distinct resulting clinical outcomes between HBV variants. In addition, the role of HBxAg mutations in ER stress remains unknown. However, it has been reported that HBxAg itself could exert ER stress in infected cells, resulting in HCC generation in chronic HBV patients. To date, review papers regarding ER stress-mediated HBV mutation have been limited into a specific mutation type: preS2 deletion. So, in this review, we will discuss details about various mutation types in all four regions of the HBV genome (preS1, preS2, S, and C) related to ER stress and their distinct ER stress mechanisms and clinical outcomes in terms of mutation types.

## 1. Introduction

Hepatitis B virus (HBV) infection is responsible for the chronic infection of more than 240 million people worldwide [1,2]. It has been estimated that more than 887,000 people die every year due to serious liver diseases, including liver cirrhosis and hepatocellular carcinoma (HCC) [3]. Despite the clear relationship between HBV infection and the development of liver cirrhosis and HCC, the underlying mechanism is still not fully known.

HBV is a member of the Hepadnaviridae family and is a small enveloped DNA virus with a virion diameter of 42 nm containing four overlapping open reading frames (ORFs) named C, S, P, and X. These encode core proteins that form viral capsids; small (SHBs), medium (MHBs), and large (LHBs) envelope proteins that form subviral particles; a polymerase; and the HBx protein, respectively [4]. Of the three proteins encoded by the S ORF, HBsAg is produced at a high rate and can be secreted independently of LHBs and MHBs. Overexpression of LHBs blocks HBsAg secretion [5]. Proper stoichiometry between LHBs and HBsAg envelope proteins is important for the secretion of SHBs and virions [6,7].

Because large amounts of proteins and lipids are synthesized in the liver, hepatocytes have a well-functioning endoplasmic reticulum (ER) with appropriate adaptive capability [8]. However, when ER stress is sustained due to chronic virus infection, it can cause hepatic dysfunctions that can lead to the progression of liver diseases [9]. Many studies have reported that HBV-infected cells showed large amounts of proteins in the ER, which led to ER stress and further contributed to the progression of severe liver diseases such as cirrhosis and HCC [10,11,12]. In most cases, those turned out to be mutant HBV proteins in the cells. To date, details about various mutation types in four HBV ORFs related to ER stress and those distinct mechanisms or specific clinical outcomes have rarely been reviewed. So, in this review, we will discuss details about mutation types in all four regions of the HBV genome (preS1, preS2, S, and C) associated with ER stress and their distinct ER stress-inducing mechanisms and clinical outcomes (Figure 1).

## 2. ER Stress

The ER is the intracellular organelle in charge of synthesizing, folding, trafficking, and maturing proteins, and it builds a closely connected network with other intracellular organelles such as mitochondria, the Golgi apparatus, endosomes, peroxisomes, and the plasma membrane [25,26,27,28]. Of the ER functions, the proper folding and modification of proteins are the most important roles, and only correctly processed proteins are exported to the Golgi apparatus, with misfolded proteins retained in the ER to be processed further or degraded [29,30].

In a normal situation, the ER maintains a proper balance between the influx of unfolded proteins and its folding capacity [31]. A signal transduction pathway has evolved that adapts to the new folding demands that occur when the balance is disturbed, and the responses are important in ER-rich cells such as lymphocytes and hepatocytes [32]. Viral infection gives the ER an overload of protein by synthesizing a large amount of viral proteins [33,34,35]. HBV is an infrequent virus identified to induce ER stress by established in vivo data [36]. The unfolded or misfolded proteins accumulate and aggregate in the ER, leading to ER stress [37,38]. The unfolded protein response (UPR) is an adaptive system of the ER that relieves ER stress and maintains its homeostasis [25,39,40]. However, overwhelming ER stress and following UPR can cause hepatic inflammation, cell death, tissue damage, and fibrosis, contributing to development of various liver diseases [8,41]. Furthermore, there is growing evidence that ER stress-mediated hepatic inflammation could facilitate the development of liver steatosis and injury [9,42]. Zhang et al. (2011) also reported that tunicamycin, a strong ER stress inducer caused ER stress-mediated liver steatosis and lipogenesis, when injected into mice [43].

Meanwhile, it is now well known that various genotypes of HBV promote development of liver disease with distinct severity. Particularly, genotype C has been repeatedly reported to show worse clinical outcomes. In a meta-analysis research with 43 studies including 14,545 patients, 25% of patients infected with genotype C strain had HCC while only 12% of patients infected with genotype B strain had HCC [44]. PreS/S deletions has been considered as an independent risk factor for HCC and progression of liver diseases, and several studies reported that genotype C more frequently showed preS/S deletions than genotype B [45,46]. Consistently, Somenath Datta et al. (2014) compared serum and tissue samples of LC and HCC patients from India, and they found that genotype C showed higher reactive oxygen species and ER stress level compared with those of genotype B [47]. Sugiyama et al. (2006) also analyzed different HBV genotypes (Aa/A1, Ae/A2, Ba/B2, Bj/B1, C, and D) and those distinct clinical criteria, and they reported that the level of GRP78, a marker of ER stress, was significantly higher in genotype C and Ba than the other geno- or subgenotypes (*p* < 0.01) [48]. Meanwhile, a recent study performed in Africa region, Bhoola and Kramvis (2016) reported that genotypes A is mainly found in southern Africa with subgenotype A1 predominating. They found that genotype A1 showed higher ER stress with overactivation of ER stress transducers compared with genotypes A2 and D3 via in vitro model experiments [49].

There are three UPR transducers in the ER: PRKR-like endoplasmic reticulum kinase (PERK), inositol-requiring protein 1 (IRE1), and activating transcription factor 6 (ATF6) [50].

PERK is a transmembrane protein located in the ER. In its normal state, PERK is bound with the ER chaperone glucose-regulated protein (GRP78/BiP) [51]. As ER stress occurs, BiP is dissociated from PERK, and PERK changes its oligomerization state from a monomer to an oligomer as it is activated [52]. Once activated, PERK phosphorylates eIF2a, which is associated with the attenuation of translation/protein synthesis. Upon the phosphorylation of eIF2a, activating transcription factor 4 (ATF4) mRNA is translated, and UPR target genes related to apoptosis are induced [53].

IRE1 is a type I transmembrane protein that has dual functions as a kinase and an endoribonuclease [51,54]. Under stress conditions, GRP78 is sequestered to misfolded or unfolded proteins in the ER, and then IRE1 is released. Its endoribonuclease (cytosolic RNAse domain) generates a transcription activator called XBP1 to increase protein folding capacity or lead to the transcriptional induction of genes encoding protein degradation enzymes [55]. Simultaneously, the kinase activity of IRE1 induces apoptotic signaling kinase-1 (ASK-1), Jun-N-terminal kinase (JNK), and p38 mitogen-activated protein kinase (p38 MAPK), leading to apoptosis [56,57,58].

ATF6 is a type II transmembrane protein in which the N-terminus is in the cytoplasm [59,60]. Similar to PERK and IRE1a, ATF6 is covered by GRP78, but in the presence of ER stress, ATF6 is released and translocated to the Golgi apparatus. After it is cleaved by S1P and S2P in the Golgi, the functional part of ATF6 is sent to the cytosol and then to the nucleus [61]. In the nucleus, ATF6 induces ERAD proteins and chaperones [62].

## 3. Mutations in the HBV PreS1/S2 Region Related to ER stress

### 3.1. PreS1 and PreS2 Mutation Type Relate to ER Stress

Many studies have consistently reported the relationship between preS mutants (LHBs and MHBs) and liver disease progression, and ER stress is considered to be a key core underlying mechanism for liver disease progression [63]. Both preS1 and preS2 mutants have defective capacity to secrete surface proteins, and this accumulation in the ER causes ground glass hepatocyte (GGH) formation, which is a histological symbol of chronic hepatitis B infection.

#### 3.1.1. Human Sample

Wang et al. (2003) investigated a total of 50 samples from eight resected liver specimens to verify the types of GGHs harboring preS1 and preS2 mutations and discovered the activation of ER stress by different preS mutants via an in vitro model [13]. They found that type I GGHs contained deletions in the preS1 region as well as an inclusion-like pattern of hepatitis B surface antigens, while type II GGHs had deletions over the preS2 region, particularly a cytotoxic T cell binding epitope region.

Pollicino et al. (2012) [16] screened 40 untreated CHB patients, and among them, 14 patients (35%) showed single or multiple preS or S mutations. They found a negative correlation between preS/S mutations and secreted HBV surface antigen levels (*p* = 0.005), that envelope proteins were restricted to the ER and that extracellular virion levels in mutants were lower than those in the WT. In addition, preS/S mutants had significantly increased cccDNA levels in the nucleus compared with those of the WT. The preS/S mutants showed a close association with male gender (*p* = 0.021) and the development of cirrhosis (*p* = 0.014).

Wang et al. (2006) [17] investigated serum and cancer tissues of HCC patients and reported that 60% of HCC patients had preS mutations in the preS1 region, preS2 region, or both regions. The mutants usually have shorter LHBs due to internal deletions. They reported that preS2-mutant LHBs prevail over preS1-mutants in the serum and tissue of HCC patients.

#### 3.1.2. Cell Lines

In a recent study, Montalbano et al. (2016) [11] demonstrated the activation of ER stress signals in transfected hepatocytes (Huh7 and HepG2 cells) with HBV-containing mutant envelope proteins (pSVL plasmid with a mutation in preS1 and pSVM plasmids with a mutation in preS2). Immunofluorescence microscopy clearly showed that LHBs localized to the ER in both cell lines and that only partial MHBs colocalized to the ER lumen. In addition, there are many studies reporting the induction of ER stress caused by LHBs and related liver disease progression. Xu et al. (1997) [14] found that overexpressed LHBs accumulated in the ER lumen and that their retention led to ER stress. They reported that an appropriate ratio between LHBs and HBsAg is necessary for the secretion of HBsAg and virions. In this regard, Ou and Rutter [64] also found that the overexpression of LHBs containing preS1 obstructed the secretion of HBsAg.

Bock et al. (1999) [15] examined three naturally occurring mutations in the preS region, MUT1 (a point mutation in the CCAAT box), MUT2 (a 6 bp deletion 38 of the CCAAT box), and MUT3 (a 153 bp deletion in the preS2 region). They found the accumulation of the mutant S proteins in the ER. MUT1 and MUT2, or a combination of MUT1 and others, showed significantly decreased extracellular S protein (60–75% reduction) compared with that of the WT; while MUT3 was related to abnormalities in viral particles. Collectively, preS mutations affect S gene modulation and lead to the development of liver disease.

#### 3.1.3. Animal Model

Chisari et al. (1987) reported that a transgenic mouse model overexpressing LHBs showed distended HBsAg in the ER [65]. As mice were treated with LPS or γ IFN, severe acute liver disease occurred, but other littermates, which were not transgenic mice, did not develop acute liver disease, suggesting the role of mutant envelope proteins in host liver damage via ER stress.

Hildt et al. (2002) also found that truncated MHBs (MHBs^t^) were not secreted but retained in the ER, while normal MHBs, which were full-length, were not retained in the ER. Cytoplasmic proteins can approach the cytosolic region of the mutant preS2 retained in the ER, leading to the activation of transcription factors [66]. In transgenic mice expressing preS2-MHBs^t^, c-Raf-1/Erk2 signaling was activated, leading to an elevation of the hepatocyte proliferation rate. In transgenic mice older than 15 months, the occurrence of liver tumors increased. These data indicated that MHBs^t^ can act as a tumor promoter and activate proliferation.

### 3.2. ER Stress Inducing Mechanism by Mutants in PreS1 and PreS2 Regions

In an in vitro study performed by Wang et al. (2003), Huh7 cells were transiently transfected with mutant preS antigens (preS1 and preS2) and signals related to ER stress were analyzed by Northern blot, Western blot, and RT-PCR assays. As shown by Northern blot analysis, the preS1 mutant significantly activated GRP78 and GRP94, six- and two-fold, respectively, compared with the WT. Meanwhile, the preS2 mutant showed lower expression of GRP78 and GRP94 than the mutant preS1 antigen. In addition, Western blot data showed that the mutant preS1 antigen dramatically enhanced the phosphorylation of JNK, which was not shown in the preS2 mutant. Then, the GRP78 gene expression level was analyzed in laser capture microdissection (LCM)-harvested samples, and both preS1 and preS2 mutants showed significantly higher expression levels of GRP78, suggesting ER stress induction by preS1 and preS2 mutants in liver tissue.

In addition, Montalbano et al. (2016) reported that the pSVL and pSVM plasmids elevated the expression of GRP78 and CHOP in the Huh7 cell line, although IRE1a and ATF4 levels were not significantly changed. Consistent with this, immunofluorescence data revealed increased GRP78 levels, which were similar to those with thapsigargin (TG), a positive control. Additionally, Huh7 cells transfected with pSVM showed a 2.0-fold increase in GRP78 expression levels, while transfection with pSVL showed no significant change in expression level. Furthermore, both pSVM- and pSML-transfected Huh7 cells revealed XBP-1 splicing, which was compared to that of the positive control, TG-treated cells. Not only was the IRE1a/XBP-1 arm activated, but the PERK axis was also activated by pSVL and pSVM, and the phosphorylation of PERK and eIF2a was highly increased. These data collectively suggest that the mutant HBV envelope proteins activate the ER stress pathway in hepatocytes [11].

According to the study of Wang et al. [17], both preS1 and preS2 mutants were retained in the ER and subsequently induced ER stress while the ER stress chaperones GRP78 and GRP94 were induced. Furthermore, Huh7 cells having preS mutations exhibited increased reactive oxygen species (ROS) and oxidative DNA damage, which were shown in transgenic mice expressing preS mutant LHBs and GGH. The oxidative DNA damage due to the preS mutations may cause genomic instability, finally leading to HCC. Huh7 cells carrying the preS mutant LHBs exhibit enhanced levels of ROS and oxidative DNA damage (also in transgenic mice carrying mutant preS LHBs and GGH). In addition, they detected that preS mutants upregulated COX-2 and cyclin A, which are associated with the induction of cell cycle progression, particularly in many types of cancer cells. Liver and kidney tissues from transgenic mice with preS mutations showed increased expression levels of COX2 protein. Consistently, in human HCC, samples expressing preS mutants had elevated COX2 mRNA levels. Considering that the transcription factor NFκB is required to activate the COX2 promoter, they treated samples with tunicamycin or brefeldin A—well-known ER stress inducers—and observed that NFκB, especially its p65 subunit, was translocated to the nucleus. Similarly, treatment with an NFκB inhibitor abrogated the induction of COX2, which was activated by both preS mutations and ER stress inducers. In addition, the inhibition of p38 MAPK blocked NFκB DNA binding and reduced COX2 induction, suggesting that preS mutations may induce COX-2 via NFκB and p38 MAPK. Furthermore, transgenic mice expressing preS2-LHBs showed hepatocyte dysplasia and tumor formation, suggesting a strong association between preS mutations and the development of HCC.

### 3.3. Clinical Implication of ER Stress Related preS1/S2 Mutations

Many studies reported that pre-S deletion has positive relationships with liver disease progression, particularly HCC [67,68,69,70]. It was also revealed that deletion in the C-terminal half of the preS1 region was frequently found among the CH and LC patients, while deletion in the preS2 region was considered a prime trait of HCC patients [67,68]. In addition, Lin et al. (2017) [71] recently reported the associations of different types of preS deletions with CH and LC. Of note, they reported that one type of preS1 deletion mutant with deletion in the C-terminal half of the preS1 region via RNA splicing was shown with very high frequency in CH (85.7% vs. 25.0%) and LC (86.4% vs. 25%) patients compared with that in ASCs. There is also some evidence supporting positive relationship between preS1/S2 mutants and childhood HCC. Abe et al. (2009) [72] found that HBV preS deletion was prevalently detected in 27 of 30 (90%) child patients with HCC. Among the mutants, preS2 deletion (74%) was most frequent followed by preS1 (18.5%) and both preS1/S2 mutant (7.4%). They assumed that HBV preS2 deletion mutant at nucleotide 4-57 containing a CD8 T-cell epitope could attribute to the development of childhood HCC. Similarly, Huang et al. (2010) also found that the higher prevalence of the mutants in children patients with HCC versus those with CH [73]. Taken together, the MHBs and LHBs variants due to mutations in the preS region could lead to ER stress in infected hepatocytes via their increased intracellular accumulation or increased HBV surface protein accumulation, mainly due to the incorrect ratio between LHBs and SHBs, thereby contributing to HCC generation in chronic HBV patients. The preS mutation patterns and their roles in HCC progression are summarized in Table 1 and Figure 2.

## 4. Mutations in the HBsAg Region Related to ER Stress

### 4.1. HBsAg Mutation Type Relate to ER Stress

HBV small surface antigen (HBsAg) has two domains: a T-cell epitope coinciding region (amino acids 28–51) [74] and another (amino acids 124–148) containing a region corresponding to the B cell epitope (“a” determinant) [20,75]. Khan et al. (2004) reported that naturally occurring mutations in the “a” determinant in S gene were located in the ER lumen and showed decreased HBsAg or HBV virion levels [76].

In our previous study, we introduced a total of 10 types of HBsAg variants from a Korean occult cohort that could be divided into five groups according to intracellular and secretion levels of HBsAg and virions [77]. In our further study of these variants, we investigated an occult infection-related HBsAg variant (KD variant with five mutations: W36L, T47K, N52D, V184A, and F220L), which belonged to group II and showed a high level of intracellular HBsAg due to an almost negative secretion capacity [18].

A Taiwanese group reported two naturally occurring mutations (W74L and L77R) in the small surface protein of HBV [19]. Huh7 cells transiently transfected with W77R showed 2.8-fold and 10-fold reductions in extracellular HBsAg and virion levels, respectively.

### 4.2. ER Stress Inducing Mechanism by the Mutants

In our previous study the KD variant, in transiently transfected Huh7 cells, showed significantly higher colocalization coefficients with the ER than WT HBsAg, while its secretion capacity was substantially reduced compared to that of WT HBsAg. In addition, the ER stress-related proteins IRE1a, ATF6, PERK, eIF2, XBP1, CHOP, and GRP78 were significantly upregulated in the KD variant. ER stress caused the release of calcium from the ER to the cytoplasm, which induced intracellular reactive oxygen species (ROS). Interestingly, we also found that the expression of antioxidant proteins such as superoxide dismutase (MnSOD and CuZnSOD), HO-1, and catalase were substantially reduced in the KD variant [18]. These data strongly implied the pathway of ER stress-linked ROS generation [78]. Furthermore, the activation of apoptotic factors was obviously shown in the KD variant; for example, the phosphorylated form of JNK, which is associated with ER stress-linked apoptosis, was highly expressed, and bax, a proapoptotic protein, was upregulated, while bcl-2, an antiapoptotic protein, was downregulated. Consistent with this, the KD variant upregulated caspase 3, 9, and 12, which facilitate apoptotic cell death, which was supported by cell death ELISA and FACS data. These data suggest that there may be a positive correlation between occult infection-related HBsAg variants and the progression of ER stress-mediated liver disease.

Chua et al. (2005) found that mutant HBsAg was retained in the ER and Golgi apparatus, whereas WT HBsAg was diffuse across the cytoplasm via immunofluorescence microscopic examination [19]. Their Western blot data showed a 10-fold elevated level of intracellular mutant HBsAg compared with that of the WT. However, interestingly, GRP78 mRNA and protein levels were not increased, although the localization of L77R HBsAg was restricted to the ER and Golgi apparatus. They presumed that the retention of L77R HBsAg in the ER is not as strong of an ER stress inducer, as shown in LHBs of HBV [65].

### 4.3. Clinical Implication of ER Stress Related HBsAg Mutations

There is some evidence suggesting that occult HBV infection (OBI), defined as the presence of HBV DNA in liver tissue or serum of individuals but negative testing for serum HBsAg, could be positively related to development of HCC in CHB patients [79]. Our previous report that various types of HBsAg mutants showing secretion defective phenotype [77] were detected from Korean individuals with OBI suggests potential linking between OBI, ER stress and HCC development. Taken together, HBsAg variants, particularly occult infection-related variants, could lead to ER stress in infected hepatocytes via their increased intracellular accumulation, potentially contributing to HCC generation in chronic hepatitis B patients. The HBsAg mutation patterns and their roles in HCC progression are summarized in Table 1.

## 5. Mutations of the HBV Core Region Related to ER Stress

### 5.1. HBcAg Mutation Type

In our previous study, a total of 70 Korean chronic hepatitis B patients with genotype C were screened, and several mutations in the precore/core region of HBV were detected [21]. Among the naturally occurring mutations, P5T/H/L in HBcAg caused ER stress and led to the production of ROS, higher intracellular calcium concentrations, the induction of inflammatory cytokines, and apoptosis.

### 5.2. Induction of ER Stress by HBcAg Mutants and the Underlying Mechanism

Huh7.5 cells transiently transfected with the three different HBV core P5T/H/L mutants showed significantly increased expression levels of GRP78. P5T and P5H mutations elevated the phosphorylation level of PERK and eIF2a. Meanwhile, P5L and P5H mutations showed a high colocalization coefficient with the ER. ER stress-related proteins, including ATF6, GRP78, pPERK, peIF2α, and pIRE1α, were significantly increased when Huh7 cells were transfected with the HBcAg mutants. In particular, the P5H mutant showed highly increased levels of ATF6, GRP78, and cytochrome c, which play a role in apoptosis, and elevated intracellular ROS and calcium concentrations in the cell. In addition, P5L and P5H mutations significantly increased the activation of NFκB as well as the mRNA levels of IL-6 and TGF-β, which are closely associated with fibrosis and HCC progression. Notably, HBcAg variants led to increased HBsAg secretion in infected hepatocytes in contrast to that in mutants of the preS/S region, which always showed defective HBsAg secretion. The effect of the P5H mutation on HBsAg secretion was further confirmed using an in vivo hydrodynamic injection model. This suggests that there may be a distinct mechanism related to ER stress induction between HBcAg and preS/S variants. In a previous study, the idea that the ER stress response could increase the synthesis and secretion of HBsAg was investigated [17]. Therefore, the P5T/H/L variants of HBcAg could induce ER stress in infected hepatocytes via ER accumulation of HBcAg, leading to enhanced HBsAg synthesis and secretion via ER stress-mediated S promoter activation. Given the role of HBsAg as an immune evasion mechanism of HBV, it is tempting to speculate that the increase in HBsAg caused by the HBcAg variant may contribute to disease progression by persistent HBV infection. The role of the P5T/H/L variants of HBcAg in HCC generation is also summarized in Table 1 and Figure 2.

### 5.3. Clinical Implication of ER Stress Related HBcAg Mutations

Previously, we have reported that mutations in preC/C region were associated with clinical severity [80]. In this report, particularly, MHC class II restricted region (M2RR) was frequently found in HCC patients (2.7% vs. 1.9%, *p* = 0.024) with subgenotype C2. Furthermore, in patients infected with genotype A1, HBcAg mutations were also most highly found in the M2RR. In addition, of M2RR mutations, P5H/L/T or P5R, mutations in residue 5 of HBcAg, which also proved to lead to hepatic ER stress, have been reported to be significantly associated with HCC patients in Indian patients infected with genotypes A or D [81] as well as in patients infected with subgenotype C2. Taken together, naturally occurring HBcAg mutations such as P5H/L/T generated particularly via immune pressure of CD4 T cell response, could contribute into HCC development in CHB patients via inducing ER stress in infected hepatocytes via the ER accumulation of HBcAg, which could in turn lead to hepatic inflammation and enhanced HBsAg synthesis and secretion.

## 6. Role of HBxAg in ER Stress

HBx is well known to have an important role in HBV replication and the development of hepatocellular carcinogenesis [82,83,84,85,86,87]. It has been consistently reported that the HBx protein of hepatitis B virus can induce ER stress, but the underlying mechanism of this has not yet been fully elucidated. To date, several HBxAg mutation types including V5M or H94Y, related to disease progression and HCC, have been introduced, but their relationships with ER stress has not been explored.

Several studies have investigated the induction of ER stress-mediated by the HBx protein. Li et al. (2007) [22] identified the role of the HBx protein, which exerts ER stress by inducing the IRE1-XBP1 and ATF6 UPR pathways. Similarly, Cho et al. (2011) [23] investigated the molecular mechanism by which HBx triggers ER stress and its sequencing activation of cyclooxygenase 2 (COX2) expression, an important factor in inflammation. They found that the mitochondrial inner membrane potential was decreased in HBx-exerting cells, which led to metabolic dysfunction in liver cells. Metabolic dysfunction reduced intracellular ATP levels; consequently, the PERK, eIF2α, and ATF4 pathways were activated. The induced ER stress increased COX2 mRNA and protein expression levels. In addition, the induction of COX2 expression was confirmed in hepatocytes and a transgenic mouse model expressing HBx. Western blot data showed that levels of ATF4, peIF2α, and the cleaved form of ATF6 were increased in HBx transiently transfected Huh7 cells and in liver tissues from HBx transgenic mice. Particularly, they found that ATF4 enhanced the expression of COX2 by binding to the COX2 promoter. The ATF4-binding site (5′-TTACGCAATA-3′) was identified, and its binding mechanism was confirmed via ChIP assay. This study indicated that HBx-mediated COX2 induction can contribute to inflammation of the liver via ER stress and mitochondrial dysfunction.

Meanwhile, a recent study by Li et al. (2017) revealed that the HBx protein modulates ER stress response so that it can inhibit apoptosis for its own survival [24]. They found that HBx protein interacted with GRP78 directly, while it was retained in the ER lumen, and its binding to GRP78 relieved ER stress. This interaction led to a reduction in eIF2α phosphorylation and ATF4, CHOP, and Bcl-2 expression. On the other hand, IRE1α, which generates prosurvival signals, was highly activated by HBx, and consequently, the expression of spliced XBP1 was increased. ATF6 was activated at the late time point. Thus, HBx may affect IRE1α and ATF6 of the UPR for cell survival. Meanwhile, PERK, another important UPR transducer, was screened in the presence of HBx, and eIF2α and ATF4 expression was decreased or postponed. Furthermore, HBx-mediated ER stress inhibited apoptosis. The expression levels of CHOP and Bcl-2, an antiapoptotic gene regulated by CHOP, were significantly decreased by HBx. Additionally, HBx inhibited poly (ADP ribose) polymerase 1 (PARP-1), which is an important administrator of DNA repair and apoptosis. Collectively, this study indicated a new mechanism of the HBV-ER stress axis that allows escape of ER stress-induced apoptosis for survival. The HBx-induced UPR signals and their roles in HCC progression are summarized in Table 1.

## 7. Summary

It seems that HBV variants may have distinct capacity or pathway to induce ER stress and definite resulting clinical outcomes. For example, in HBsAg variants mainly related to OBI, ER stress occurred via increased intracellular accumulation of HBsAg variants. Direct ER accumulation of HBsAg mutants could lead to strong ER stress response beyond maintenance of cell homeostasis by normal UPR, resulting in liver dysfunction via apoptotic cell death. In MHBs and LHBs variants generated by mutations in the preS region, perhaps due to immune pressure or RNA splicing, ER stress occurred via increased intracellular accumulation of MHBs or LHBs variants, or via increased accumulation of HBsAg due to the incorrect ratio of produced LHBs and HBsAg. In contrast to variants of HBsAg, MHBs and LHBs showing defective HBsAg secretion, HBcAg P5T/H/L of M2RR variants generated by host CD4 T cell responses could cause ER stress via ER accumulation of the HBcAg variant, leading to enhanced secretion of HBsAg via S promoter activation. Although there are some discrepancies in ER stress-mediated biological functions according to the type of variants and their ER stress-inducing capacity, they can generally elicit ER stress-mediated biological responses such as ROS production, inflammatory cytokine production, TGF-β secretion, hepatocyte proliferation followed by apoptosis, and enhanced HBsAg secretion (in case of HBcAg variant), all of which are linked to the progression of liver disease. Therefore, prolonged inflammation, liver damage and increased HBsAg secretion in these naturally occurring variants may contribute to the progression of liver disease.

## 8. Conclusions

In conclusion, in the natural course of HBV infection, the accumulation of naturally occurring mutations in the HBV genome can generate several mutant types of HBV-encoded proteins, including three different proteins in the S ORF (SHBs, MHBs, and LHBs) and HBcAg in the C ORF, which could contribute to enhanced ER stress in infected hepatocytes mainly via increased ER accumulation of mutant proteins. The mutants may have distinct capacity and pathway to induce ER stress and result in definite clinical outcomes. In addition, the role of HBxAg mutations in ER stress remains still unknown. However, it has been reported that HBxAg itself could exert ER stress in infected cells, resulting in HCC generation in chronic HBV patients.

## Figures and Tables

**Figure 1 ijms-20-00597-f001:**
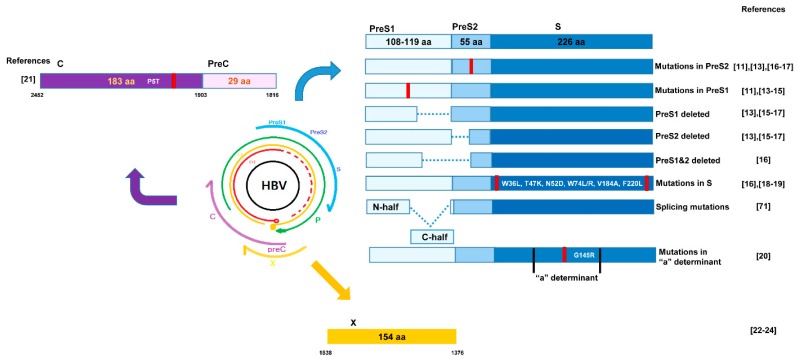
Locations and types of hepatitis B virus (HBV) mutations reported to induce endoplasmic reticulum (ER) stress in previous literature. Mutations in preS1 ([11,13,14,15]), and PreS2 ([11,13,16,17]) deletion in PreS1 and preS2 ([13,15,16,17]), or deletion in both region ([16]) are indicated. Several mutations in S regions are reported ([16,18,19]) including those in "a" determinant ([20]). Mutations in core ([21]) and X region ([22,23,24]) reported to cause ER stress are indicated.

**Figure 2 ijms-20-00597-f002:**
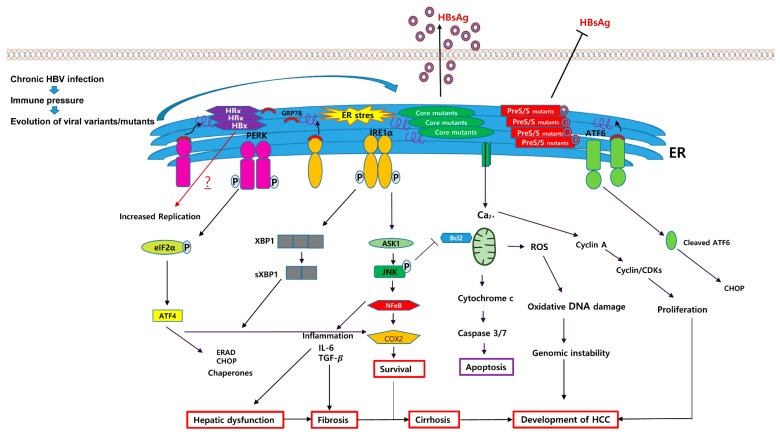
Schematic pathway for ER stress-mediated HCC generation caused by several types of HBV variants such as preS/S, core mutants, and HBx proteins. Naturally occurring mutant HBV proteins, which are caused by immune pressure during chronic HBV infection, including HBsAg, MHBs, LHBs, and HBcAg, could accumulate in the ER and induce ER stress. The HBsAg, MHBs, and LHBs variants could lead to defective HBsAg secretion due to disturbance of the ratio between LHBs and HBsAg. Meanwhile, HBcAg variants could increase the capacity of HBsAg secretion, which may be due to activation of the S promoter via ER stress, and induced ER stress could activate three different UPR transducers: PERK, IRE1, and ATF6. Generally, ER stress-mediated UPRs could contribute to HCC generation in chronic HBV patients via distinct biological actions including ROS production, inflammation, apoptosis, and the activation of NF-κB. Black arrows indicate positive regulation. T-bars indicate suppression. The red arrow implies an unknown consequence of ER stress yet via HBV genome mutation for the progression of HCC. HBsAg, hepatitis B virus surface antigen; MHBs, hepatitis B middle surface protein; LHBs, hepatitis B large surface proteins; HBcAg, hepatitis B core antigen.

**Table 1 ijms-20-00597-t001:** Naturally occurring mutations in HBV leading to endoplasmic reticulum (ER) stress and their contribution to hepatocellular carcinoma (HCC).

Regions	Types	Localization	Extracellular HBsAg	Extracellular HBV Virion	Unfolded Protein Response (UPR) Signals	HCC-Related Factors	Apoptosis	Reference	Genotype
**PreS1**	PreS1 mutants with deletions	ER	Reduction		Upregulation of PERK, c-JNK, GRP78, GRP94			[13]	
Overexpressing LHBs	ER	Reduction		Activation of IRE1, ATF4, CHOP, PERK, c-JNK, GRP94, sXBP-1			[11]	
Overexpressing LHBs	ER	Reduction	Reduction	Activation of GRP78 and GRP94			[14]	
Deletion in PreS1	ER	Reduction	Reduction				[16]	A and D
Deletion in PreS1	ER	Reduction	Reduction/abnormal	Activation of GRP78 and GRP94	COX2, Cyclin A, NFκB, ROS		[17]	
PreS1 single mutation	ER	Reduction	Reduction/abnormal				[15]	
**PreS2**	overexpressing MHBs	ER (partially)	Reduction		Activation of IRE1, ATF4, CHOP, PERK, c-JNK, GRP78, GRP94, sXBP-1			[11]	
Point mutation /deletion abolished start codon	ER	Reduction	Reduction				[16]	A and D
PreS2 mutants with deletions	ER			Activation of GRP78 and GRP94	COX2, Cyclin A, NFκB, ROS		[17]	
Truncated MHBs	ER	Reduction			c-Raf-1/Erk2 signaling, Tumor growth	Upregulation	[66]	
Deletion in the preS2	ER	Reduction	Reduction/abnormal				[15]	
**S**	W36L, T47K, N52D, V184A, F220L	ER	Reduction		upregulation of IRE1, ATF6, PERK, eIF2, XBP1, CHOP, and GRP78		Upregulation	[18]	C2
W74L and L77R	ER and Golgi apparatus	Reduction		no increase in GRP78 mRNA and protein (ER stress response)			[19]	
**Core**	P5T/H/L	ER	Increase (serum/transgenic mice)		Activation of, ATF6, GRP78, pPERK, peIF2α, and pIRE1α	ROS, inflammatory cytokines, cytochrome c, NFκB, TGF-β	Upregulation	[21]	C2
**HBx**	pcDNA-HBx				Induction of GRP78, IRE1-XBP1, and ATF6			[22]	
HBV(pcDNA3.1)-expressing HBx				Activation of PERK, eIF2α, ATF4, sXBP-1, CREB-H	COX2, liver dysfunction, inflammation		[23]	
The pLV-cDNA containing HBX gene	ER			Interaction with GRP78 (binding), downregulation of p-eIF2α ATF4, CHOP, Bcl-2	Avoiding the activation of ATF4-mediated DNA repair	Inhibition	[24]	
				Activation of IRE1a, sXBp-1, and ATF6

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
