# Peer review of "Naturally Occurring Hepatitis B Virus Mutations Leading to Endoplasmic Reticulum Stress and Their Contribution to the Progression of Hepatocellular Carcinoma"

_ijms, 2019, doi:10.3390/ijms20030597_

Reviewer 1 Report

The aim of this review was to discuss the mutations in all four regions of the HBV genome (preS1, preS2, S and C) that have been reported to induce ER stress and their underlying ER-stress-inducing mechanisms.
The work is interesting, and may carry practical importance. However, I have some comments that could improve the paper:
In general, not all of the information is logically combined into  a meaningful thread.
1.      The part 1 and 2 (“Introduction and ER Stress”)  are too general; I recommend the authors give more background  and make a good transition between the background and the aim  of the study. How is this work different from others and what is the significance of the work? There is too many references concerning to general knowledge about the endoplasmic reticulum (ER) (14-190. Between lines 63 and 64 I recommend add some sentences (or more data) that the unfolded protein response (UPR) is the main mechanism of damage by HBV. In this fragment the reader should know why UPR transducers are described in details.
2.      I recommend also to check precise all definitions  of mutations, as used in Table 1 and on Fig. 1, e.g.   “Khan et al. (47) have identified in their study four  naturally occurring mutations in the S domain that enhance  or impair virion secretion”; is better to change in line 99  ?a determinant” to ?a determinant in S gene”.
3.      I recommend to arrange the chapters/subchapters of the review as in Table 1, it will be easier for the reader.
4.      In current work I lack the description of the  exact mechanisms and clinical implications/effects of  HBV mutations. I recommend supplementing this data with  the clinical consequences of ER damage in patients with HBV mutations. Are any connections between different HBV mutation and liver steatosis via ER stress? Could you describe more precisely the connection between the HBV mutations and PROGRESSION, not only the general development of HCC?
5.      I recommend to underline on Figure 2 so far  unknown consequences of ER stress via HBV genome  mutations for the progression of HCC, according the title of the paper.
6.      I suggest to shorten the description of some data, e.g. Wang et al. [58], the description takes many lines (188-210).
7.      I suggest to change some older work on epidemiology with chronic HBV infection with newer literature –ref. 1-8.
8.      At the end of the review, I recommend the  authors have a summarizing section to describe the  authors’s opinions regarding the significance of the  processe underlying naturally occuring HBV mutation, to answer if the review of different studies give new concepts for the implications for understanding and managing drug resistance, immune escape, vaccination, pathogenesis, viral mutation rates, etc.
9.      The linguistic style or mistakes needs some  improvements:a.      L78 change ?Yoshida H, 2001” to the number  of ref.b.      L83 and in Table 1 – change ?Golgi” to  ?Golgi apparatus” or ?Golgi complex”c.       In most of the references lack the  names of scientific journals; some other details  of references list should also be corrected according  the journal requirements.

Author Response

Dear, Dr. Reviewer 1
Hi, Sir/Ma’am,
Above all things, thank you for your great advice and help that make our review paper to be more organized and educational. We really appreciate you for your thoughtful considerations.
As you recommended,
1-2) We added more specific background/information regarding ER stress in the part 1 and 2 (introduction and ER stress) so that readers may understand the transition from the background to the information about UPR with no difficulties. And we tried to specify the words/definitions used in the script and revised the part you recommended to change such as “a determinant” to “a” determinant in S gene (Line 221)
3) We arranged the chapters and sub-chapters in our review paper based on the sections in Table 1.
Plus, we specified more subsections according to another reviewer’s opinion (e.g. cell lines, animal studies, and human samples, etc.)
4) According to your suggestions, we add supplementing clinical implications/effects of HBV mutations including steatosis, chronic hepatitis, Liver cirrhosis, and HCC developed via ER stress. We supplemented the information about specific mutation types which are closely related to distinct clinical consequences including CH, LC, and HCC (not just to HCC).
5) Also, according to your recommendation, we underlined an unknown consequence of ER stress via HBV genome mutation (HBx) for the progression of HCC (Figure 2).
6) As you suggested, we’ve shorten some data and break down the contents into small groups.
7) We altered some older references to the latest ones regarding epidemiology with chronic HBV infection (except for some historical discover such as ref. 5 or 6) and other references throughout the paper.
8) According to your advice, we spared some space at the end of the paper for summarization. We described the authors’ opinions regarding the importance of the naturally occurring HBV mutations and those underlying mechanisms leading to severe liver diseases.
9) Some minor errors were corrected regarding reference format (including journals’ name) and word usage (precise).
Again, thank you very much for your help and suggestions.

Sincerely,

Yumin Choi

Department of Microbiology and Immunology, Biomedical Sciences, Liver Research Institute and Cancer Research Institute, College of Medicine, Seoul National University, Seoul, Korea

Reviewer 2 Report

The authors have taken up a difficult topic.  The work unambiguously determines the impact of HBV mutations on the development of HCC as it is specified in the title.
The aim of manuscript is not clearly explained, e.g. what's new in this problem and why the authors decided to focus on this problem.
There is no clinical information in text:
- about frequency of listed mutants in different stages of HBV infection,
- no shown differences between HBV genotypes, between  adults and children.
The authors did not focus on the differences in the occurrence of mutations in the course of chronic hepatitis, liver cirrhosis and HCC.
There is no information on splicing mutations in the analyzed regions.
The manuscript lacks a clear division into research done on cell lines, in animals and humans
Recently, many works have been published on this topic. Nearly 85% of references are older than 5 years. It seems that when the manuscript is prepared, new published data should be cited.

Author Response

Dear, Dr. Reviewer 2
Hi, Sir/Ma’am,
First of all, thank you for your helpful suggestions and advices that make our review paper to be more informational and easy to read. We are very thankful for your considerations on our paper.
As you recommended,
- We tried to specify the impact of HBV mutations precisely on the development of liver diseases and answer what are the differences of our paper from other papers that have published on this topic.
- We added supplementing clinical information:
-       About prevalence of the mutants in different stages of HBV infections (ACS, CH, LC, and HCC)
-       Reported distinct clinical outcomes among HBV genotypes and comparison between adults and children
-       Information regarding splicing mutations in the contents and figure 1
-       Division of the data into several subsections, cell lines, in animals and human
-       Add subsections in the paper based on Table 1:
n  Mutation types
n  Clinical implications
n  Underlying mechanism leading to ER stress and further liver disease progression
-  We altered some older references to the latest ones regarding epidemiology with chronic HBV infection (except for some historical discover such as ref. 5 or 6) and other references throughout the paper.
Again, thank you very much for your help and suggestions.

Sincerely,

Yumin Choi

Department of Microbiology and Immunology, Biomedical Sciences, Liver Research Institute and Cancer Research Institute, College of Medicine, Seoul National University, Seoul, Korea

Reviewer 3 Report

This review summarized the current knowledge about ER stress caused by naturally occurring HBV mutations and their contribution to the  progression of HCC. This article is well organized and worth publication. However, before it can be accepted for publication, some edits are needed as listed below.
Major comment:
1) References: Journal names are missing in most of the references.
Minor comments:
2) Line 40: "SHBs" , same with HBsAg?
3) Figure 1.:It is hard to read " 183aa"(written black in purple), " 226aa", " G145R" and "W36L, T47K, N52D, V184A, F220L "(written black in blue).
4) Line 78: Where is the reference (Yoshida H,2001) ?

Author Response

Dear, Dr. Reviewer 3

Hi, Sir/Ma’am,

First of all, thank you for your suggestion and help that make our review paper to be more educational. We appreciate you for your positive considerations.
As you recommended,
1) We corrected the reference format and included journals’ name.
2) Line 39: changed “HBsAg” to “SHB”
(Although it depends on each paper and authors' decision, HBsAg and SHB were used as same meaning in many papers. More precise expression would be small HBV surface antigen, S-HBsAg. In the paper that I cited, HBsAg was used as same meaning of SHB so I used the word, but as you recommended or for consistency, I corrected it to SHB or surface HBV antigen/protein.
3) As you recommended, I adjusted the font size and color in Figure 1 in order to make it easy to read.
4) Line 97: We added a corresponding reference for the citation (Yoshida H, 2001)
Again, thank you very much for your help and suggestions.

Sincerely,

Yumin Choi

Department of Microbiology and Immunology, Biomedical Sciences, Liver Research Institute and Cancer Research Institute, College of Medicine, Seoul National University, Seoul, Korea

 Round  2

Reviewer 1 Report

The authors exhaustively responded to my doubts and questions posed by myself following reading of the first version of the paper. Most of the data or corrections have been introduced everywhere where they were required.
However, the names of the journals are still missing, e.g. references 3-12; 14-19; 24, 27-37; 43; 46-47, 51, 52, 55,56, 59, 60-71; 75-81; 83-87.
in the ref. no. 72 – the Journal abbreviation should be “World J Gastroenterol
In the description of the Figure 1 to finish the sentence, please add a full stop.

Author Response

Dear, Dr. Reviewer 1
Thank you very much for your help and suggestions.
As you recommended,
1) We corrected the references, which were lack of journal names, and confirmed the format. There was a problem in the Endnote program, so we fixed it. (including the abbreviation of WJG, World J Gastroenterol)
2) In the description of the Figure 1, we added a period at the end of the sentence.
Thank you for your help and suggestion that make our review paper to be more organized and easy to read.  We really appreciate you for your support!

Sincerely,

Yumin Choi

Department of Microbiology and Immunology, Biomedical Sciences, Liver Research Institute and Cancer Research Institute, College of Medicine, Seoul National University, Seoul, Korea

Reviewer 2 Report

The names of the journals in references are still not listed (numbers: 3, 4, 5, 6, 7, 8, 9,10,11, 12,14,15,16,17,18,19,24,27,28,29,30,31,32,33,34,35,36,37,42,43,46,47,51,52,55,56,57,59,60,61,62,63,64,65,66,67,68,69,70,71,75,76,77,78,79,80,81,83,84,85,86,87

Author Response

Dear, Dr. Reviewer 2
Thank you very much for your help and suggestions.
As you recommended,
We corrected the references, which were lack of journal names, and confirmed the format. There was a problem in the Endnote program, so we fixed it.
Thank you for your help and suggestion that make our review paper to be more organized and easy to read. We really appreciate you for your support!

Sincerely,

Yumin Choi

Department of Microbiology and Immunology, Biomedical Sciences, Liver Research Institute and Cancer Research Institute, College of Medicine, Seoul National University, Seoul, Korea

Reviewer 3 Report

Dear author,

1) Unfortunately, the reference format was not corrected adequately.

2) 3) 4) I was satisfied with the corrections.

Best wishes, 

Author Response

Dear, Dr. Reviewer 3
Thank you very much for your help and suggestions.
As you recommended,
We corrected the references, which were lack of journal names, and confirmed the format. There was a problem in the Endnote program, so we fixed it.
Thank you for your help and suggestion that make our review paper to be more organized and easy to read. We really appreciate you for your support!

Sincerely,

Yumin Choi

Department of Microbiology and Immunology, Biomedical Sciences, Liver Research Institute and Cancer Research Institute, College of Medicine, Seoul National University, Seoul, Korea